# Strategies to Optimize Treatment for Locally Advanced Rectal Cancer

**DOI:** 10.3390/cancers15010219

**Published:** 2022-12-29

**Authors:** Xiaoyu Hu, Zhuang Xue, Kewen He, Yaru Tian, Yu Chen, Mengyu Zhao, Jinming Yu, Jinbo Yue

**Affiliations:** 1Department of Oncology, Renmin Hospital of Wuhan University, Wuhan 430064, China; 2Department of Radiation Oncology, Shandong Cancer Hospital and Institute, Shandong First Medical University, Shandong Academy of Medical Science, Jinan 250117, China

**Keywords:** LARC, radiation therapy, immunotherapy, TNT, SCRT

## Abstract

**Simple Summary:**

Locally advanced rectal cancer (LARC) is a group of highly heterogeneous tumors. According to the European Society for Medical Oncology (ESMO) guidelines, the risk of recurrence can be classified by magnetic resonance imaging (MRI) as low to very high. In the era of precision medicine, final treatment decisions should be based on risk stratification. Not all LARC cases are appropriate for the predominant neoadjuvant “sandwich” strategy. Treatment modalities for LARC have dramatically evolved in recent years. Multiple clinical trials focused on optimizing strategies to improve outcome oncology outcomes and functions for LARC patients. In the context of precision medicine, treatment decisions for locally advanced rectal cancer should be based on risk stratification, molecular typing, and patient preference.

**Abstract:**

Neoadjuvant long-course concurrent chemoradiation plus surgery, followed by optional adjuvant chemotherapy, is a standard of care for locally advanced rectal cancer (LARC). However, this traditional approach has several limitations, including low pathological complete response (pCR) (10–25%), high metastasis rate (30–35%), and highly inconsistent compliance with adjuvant chemotherapy (25–75%). Treatment modalities for LARC have dramatically evolved in recent years. Multiple clinical trials have focused on optimizing strategies to achieve a win-win situation for oncologic outcomes and functions. Here, we review the latest studies into optimizing neoadjuvant treatment for LARC.

## 1. Introduction

Stage II (cT3-T4, N0) and stage III (cT1-T4, N1-N3) rectal cancers are commonly categorized as locally advanced rectal cancer (LARC). During the past three decades, due to precise staging and risk stratification of magnetic resonance imaging (MRI), the performance of neo-chemoradiotherapy (neo-CRT) and total mesorectal excision surgery (TME) has decreased five-year local recurrence rates from >25% to approximately 5% to 10% and improved sphincter preservation in patients with LARC [1,2,3]. Standard treatment for locally advanced rectal cancer generally involves neoadjuvant long-course concurrent chemoradiation therapy or short-course radiation without chemotherapy, followed by TME and optional adjuvant chemotherapy [4]. This standard approach has several limitations, including low pathological complete response (pCR) (10–25%), high metastasis rate (30–35%), and highly inconsistent compliance with adjuvant chemotherapy (25–75%). Treatment strategies are optimized on the basis of standard treatment to improve outcomes and organ function [5]. In recent years, multiple studies have explored therapeutic strategies to increase the intensity of neoadjuvant therapy, aiming to maximize tumor regression for organ function preservation, including intensified concurrent chemotherapy, increasing induction chemotherapy or consolidating chemotherapy, or using total neoadjuvant treatment (TNT) [4]. In the era of immunotherapy, several phase 2 studies have been conducted to explore the efficacy of neoadjuvant immunotherapy in mismatch repair defect (dMMR) LARC patients [6]. Due to the side effects of TME and radiotherapy (RT), non-operative modality (NOM) treatment—which includes: radiation dose escalation instead of surgery, local resection, watch and wait (WW), and omission of RT—should also be explored in some patients with LARC [7].

Here, we review these multiple clinical trials focused on optimizing strategies to effectively eradicate rectal cancer while preserving the organ itself.

## 2. Intensifying Concurrent Chemotherapy Regimen

The standard chemotherapy regimen used in many trials to this point has been fluorouracil (5-FU; NSABP-R03) or capecitabine (NSABP R-04) [8,9]. Despite the standard strategy having decreased locoregional recurrence rates, the pCR rate and the control of systemic disease need further improvement. Thus, to improve treatment outcomes, it is necessary to strengthen the concurrent chemotherapy regimen. Several studies have evaluated augmenting the standard therapy with oxaliplatin, with mixed results. Four randomized trials (NSBP R-04, STAR-01, ACCORD12, and CAO/ARO/AIO-04) have investigated the addition of oxaliplatin to 5-FU or capecitabine-based preoperative CRT [9,10,11,12,13]. The German CAO/ARO/AIO-04 trial showed 3-year DFS (75.9% vs. 71.2%, *p* = 0.03) and improved pCR (17% vs. 13%, *p* = 0.031), with no difference in R0 resection and grade 3–4 toxicity (25% vs. 21%) [12]. However, the other three trials showed that the addition of oxaliplatin did not improve local control, DFS, or OS and, furthermore, increased grade 3–4 diarrhea [10,11,14,15] (Table 1). The PETACC 6 clinical trials also investigated the role of oxaliplatin in combination with preoperative capecitabine-based CRT and postoperative capecitabine to improve DFS in LARC [14,15]. Long-term results confirmed that the addition of oxaliplatin to capecitabine plus RT did not improve DFS nor OS in the intention-to-treat population. However, post hoc analysis indicated that oxaliplatin increased DFS and OS in patients <60 years of age (Table 1). Finally, the results of the FOWARC trial showed that co-treatment with RT and modified FOLFOX6 (mFOLFOX6), a chemotherapeutic regimen consisting of folinic acid, 5-FU, and oxaliplatin, yielded a significantly higher pCR rate (27.5%) than that of the 5-FU + RT group (14.0%) or the mFOLFOX6 monotherapy group (6.6%) [16,17].

The FOWARC study differed from the others like it in a few key aspects, and this may explain the large discrepancy between its results and theirs. In the NSBP R-04, STAR-01, ACCORD12, and PETACC 6 trials, oxaliplatin was administered on a weekly basis [10,11,12,14]. It may be that a weekly treatment frequency is too high, giving patients’ bodies insufficient time to recuperate after treatment. Furthermore, in the FOWARC trial, oxaliplatin was given at a dose of 85 mg/m^2^, whereas the STAR-01, ACCORD-12, NSABP R-04, and PETACC-6 trials used lower doses in the 50–60 mg/m^2^ range [16,17]. These lower doses may be insufficient to achieve oxaliplatin’s role as a radiosensitizing agent. Beyond that, a post hoc analysis of the CAO/ARO/AIO-04 trial proposed that treatment adherence is crucial for the benefit of intensified concurrent chemotherapy. In the STAR-01, ACCORD-12, NSABP R-04 and PETACC-6 trials, the total dose of concurrent chemotherapy was markedly reduced in the experimental groups, and no improvement was reported for oncologic outcomes. Conversely, in the FOWARC and CAO/ARO/AIO-04 trials, adherence to nCRT was largely comparable in both groups. In the CAO/ARO/AIO-04 trial, pCR and DFS were significantly improved in the experimental group. The FOWARC trial also showed increased pCR due to the addition of oxaliplatin, whereas this trial revealed decreased distant metastasis, but neither demonstrated a DFS benefit [21].

According to the above research conclusions, the addition of oxaliplatin increased toxicity but did not increase efficacy. If the goal of treatment is to maximize tumor regression (i.e., to pursue pCR or tumor regression grade (TRG)), it is worth reconsidering the value of oxaliplatin as the standard systemic chemotherapy rather than simply as a radiation sensitizer (once a week, 50–60 mg/m^2^).

Irinotecan has been shown to be an effective radiosensitizer in colorectal cancer in preclinical studies. However, its evaluation in human subjects has, like oxaliplatin, produced ambiguous outcomes. Two clinical trials are of relevance here: ARISTOTLE and CinClare [19,20,22]. In the ARISTOTLE study, 564 patients were randomly divided into two groups: one treated with capecitabine-based CRT (C-CRT) and the other with irinotecan combined with capecitabine-based CRT (IrCRT). The pCR rate was 20.2% for IrCRT vs. 17.4% for C-CRT (*p* = 0.45). On the other hand, the rate of grade 3–4 gastrointestinal adverse events was 21% with IrCRT vs. 12% with C-CRT (*p* = 0.004). Moreover, patients receiving IrCRT had significantly more diarrhea (13.8% vs. 3.5%; *p* < 0.001) and neutropenia (9.8% vs. 1.1%; *p* < 0.001) [20] (Table 1).

CinClare investigated the role of irinotecan combined with capecitabine based on UGT1A1 genotyping. The wild-type allele, UGT1A1*1, is associated with normal enzyme activity; however, the UGT1A1*28 variant reduces UGT1A1-mediated inactivation of SN-38, an active metabolite of irinotecan, and is associated with the risk of myelosuppression and severe diarrhea [19,22]. As such, genotyping and toxicity monitoring are necessary to ensure patient safety when treating with irinotecan. All patients received preoperative pelvic RT concurrently with capecitabine, with or without irinotecan, followed by a course of XELIRI and XELOX. Patients subsequently underwent TME followed by five more cycles of XELOX. The pCR rate significantly improved in the XELIRI group (CapRT vs. CapIriRT, 15% vs. 30%, *p =* 0.001) [23]. Although treatment-related toxicity was increased, it was tolerable (Table 1).

Comparing the CinClare and ARISTOTLE studies, several key differences emerge, which may explain the discrepancies in their reported outcomes. The ARISTOTLE trial featured broader inclusion criteria. Patients were included in the ARISTOTLE study so long as they met the criteria for the locally advanced population. In contrast, in the CinClare study, subjects who were homozygous for the UGT1A1*28 allele were excluded from the study due to a high probability of serious adverse events. The administered dosage of irinotecan also differed between the two trials. All subjects in the ARISTOTLE study received irinotecan 60 mg/m^2^ weekly for four weeks. In the CinClare study, on the other hand, the dose administered depended on the subject’s genotype of UGT1A1*28, with the dose for wild-type patients being 80 mg/m^2^ per week and that for the UGT1A1*28 heterozygous patients being 65 mg/m^2^ per week. It is thus plausible that the higher dose used in the CinClare trial resulted in better tumor regression.

## 3. Short-Course Radiotherapy

The standard regimen of short-course radiotherapy (SC-RT) delivered as 5 Gy over 5 fractions, followed by surgery after one week, originated in Europe due to insurance restrictions [24]. Although SC-RT is less expensive than LCCRT, the Polish and TROG clinical trials revealed that the technique yielded inferior pCR and higher rates of positive circumferential radical margin (CRM) [25,26]. One way that has been hypothesized to improve the pCR rate for SC-RT is to increase the interval between RT and surgery. This hypothesis was evaluated in the Stockholm III clinical trial [27]. In it, patients were randomly assigned to receive A) 5 × 5 Gy radiation with surgery within either 1 week (SC-RT) or B) 4–8 weeks (SC-RT with delay), or C) 25 × 2 Gy radiation dose with surgery after 4–8 weeks (LCCRT). The results showed that delaying surgery after SC-RT gives similar results compared with SC-RT with immediate surgery. The results of the LCCRT treatment were similar to both SC-RT regimens, but with the added negative of substantially prolonging treatment time. However, postoperative complications were significantly reduced in the SC-RT with delay group compared to SC-RT with immediate surgery [27]. Therefore, a longer interval between SC-RT and surgery may improve patient outcomes by reducing treatment-related adverse events.

Another potential way to improve the pCR rate is to add chemotherapy to SC-RT. There were two published phase III trials that explored whether SC-RT plus chemotherapy is superior to LCRT: the STELLAR and Polish II trials [26,28]. The Polish II trial hypothesized that delaying surgery following SC-RT and adding consolidation chemotherapy would increase the radical resection rate compared with CRT alone. The preliminary results showed that acute toxicity and 3-year OS were better with SC-RT combined with chemotherapy than LCCRT. However, the recently updated long-term outcomes reported that local recurrence, distant metastasis, DFS, OS, R0 resection rate, pCR rate, and grade 3–4 grade toxicities were similar between the two groups [26].

In the STELLAR trial, a randomized, multicenter trial in China, patients were randomly assigned to receive either 5 Gy × 5 and 4 courses of CAPOX (SC-RT group) or 50 Gy in 25 fractions concurrently with capecitabine (LCRT group). TME in both groups was performed 6–8 weeks later, followed by either two or six courses of postoperative CAPOX in the SC-RT and LCRT groups, respectively. The interim analysis showed that acute toxicity and surgical complication were acceptable and comparable in both groups. At a median follow-up of 35.0 months, 3-year DFS was 64.5% and 62.3% in TNT and CRT groups, respectively (hazard ratio, 0.883; one-sided 95% CI, not applicable to 1.11; *p* < 0.001 for non-inferiority). There was no significant difference in metastasis-free survival or locoregional recurrence, but the SC-RT group had better 3-year OS than the LCRT group (86.5% vs. 75.1%, *p* = 0.033). The prevalence of acute grade III-V toxicities during preoperative treatment was 26.5% in the SC-RT group vs. 12.6% in the LCRT group (*p* < 0.001) [28]. Thus, although the exact parameters in which the benefit was observed differed between the two studies, both found that SC-RT combined with consolidation CT is not inferior to LCRT.

## 4. Total Neoadjuvant Therapy

Several trials have been undertaken to evaluate the role of adjuvant chemotherapy after neoadjuvant CRT/(SC-RT) and surgery. These have failed to improve DFS or OS. However, the interpretation of these trials is confounded by poor compliance resulting from postoperative complications, treatment-related adverse effects, and suboptimal regimen implementation [29,30]. When considering the limitations of postoperative chemotherapy, some clinical trials have assessed the role of preoperative rather than adjuvant chemotherapy, referred to as total neoadjuvant therapy (TNT) [31]. In TNT, chemotherapy is administered before or after neo-CRT and prior to TME, with the hope that this will improve compliance rates, reduce toxicity, and decrease distant relapse rates. The advantage of this approach is that, by administering chemotherapeutics prior to RT or surgery, systemic delivery of the drugs is facilitated due to the intact tumor vasculature and microenvironment. On the other hand, providing chemotherapy prior to radiosurgery can lead to a missed window of surgery, and neoadjuvant treatment toxicity precludes definitive surgical resection [32].

Two TNT paradigms have been investigated in recent trials: (1) CRT/SC-RT followed by consolidation chemotherapy, and (2) induction chemotherapy followed by CRT/SC-RT. CAO/ARO/AIO-12, a randomized phase II clinical trial, compared the efficacy of these two models of TNT. The authors found that consolidation TNT led to higher rates of pCR (25% vs. 17%), better compliance, and less grade 3/4 CRT-related toxicity (27% vs. 37%) than induction TNT [33,34] (Table 2). These findings indicate that up-front CRT followed by consolidation chemotherapy is the preferred sequence for total neoadjuvant therapy if organ preservation is a priority. The key reason may be that tumor cells in patients with consolidation chemotherapy have more time to shrink and die after radiotherapy compared to those in patients in the induction chemotherapy arm [35].

The RAPIDO study is a landmark trial of TNT aimed at investigating whether delayed surgery followed by SC-RT combined with consolidation chemotherapy can reduce the risk of systemic recurrence without affecting local control [40]. In the RAPIDO trial, participants were randomly assigned to one of two groups: the experimental and standard-of-care groups. Patients assigned to the experimental treatment group received SC-RT (5 × 5 Gy over a maximum of 8 days) followed by six cycles of CAPOX or nine cycles of FOLFOX4, as per the treating physician’s discretion or hospital policy. Thereafter, subjects were treated with TME. Patients assigned to the standard-of-care group received either A) 28 daily fractions ranging from 1.8 Gy up to 50.4 Gy or B) 25 fractions ranging from 2–50 Gy (per physician’s discretion or hospital policy), with concomitant twice-daily oral capecitabine (825 mg/m^2^) followed by TME and, if stipulated by hospital policy, adjuvant chemotherapy with eight cycles of CAPOX or 12 cycles of FOLFOX4. While the experimental TNT treatment mode had no significant effect on OS, the pCR rates of the test and control groups were 27.7% and 13.8% (*p* < 0.001), respectively. The 3-year disease-related treatment failure (DrTF) rates were 23.7% and 30.4%, respectively (*p* = 0.02). The 3-year distant metastasis rates and local recurrence rates were 19.8% vs. 26.6% (*p* = 0.004) and 8.7% vs. 6.0% (*p* = 0.10), respectively. The 3-year disease-related treatment failure and distant metastasis rates were both significantly reduced by 7%, and the pathological complete response rate was increased from 13.8% to 27.7% [40]. The RAPIDO study successfully reduced the risk of distant metastasis but did not reduce the risk of local recurrence. The biggest reason lies in the poor tolerance and low acceptance rate of postoperative adjuvant chemotherapy in the standard arm. Preoperative chemotherapy compliance and tolerance were significantly better in the experimental arm. While radiotherapy is the main factor affecting local control, the change in chemotherapy sequence has little influence on local control.

The PRODIGE 23 trial adopted a TNT-like mode to investigate the role of induction FOLFIRINOX followed by preoperative CRT, TME, and adjuvant chemotherapy. The standard-of-care group received preoperative CRT, TME, and adjuvant chemotherapy for six months. Compared with other TNT studies, the PRODIGE 23 study added irinotecan as an induction chemotherapy agent, and adjuvant chemotherapy was still added for three months after surgery. The results of the trial were as follows: three-year DFS rates in the experimental group were 75.7% vs. 68.5% in the standard-of-care group (*p* = 0.034); the pCR rate was 27.8% in the experimental group vs. 12.1% in the standard-of-care group; metastasis-free survival (MFS) in the experimental group was 78.8% vs. 71.7% in the standard-of-care group (*p* = 0.017); serious adverse events occurred in 27% of participants in the experimental group vs. 22% of patients in the standard-of-care group (*p* = 0.167) (Table 2). Altogether, these results indicate that introduction of mFOLFIRINOX before CRT is safe, significantly increases the pCR rate, DFS and MFS, and is feasible for patients with high distant metastasis [37,38]. However, whether TNT-like mode can really improve the long-term prognosis is still controversial, and whether some patients will reduce the efficacy due to overtreatment has not reached a consensus.

For patients with a high or very high risk of recurrence, the treatment goal is not only to control local recurrence but also to reduce the risk of distant metastasis, so TNT is an ideal strategy. If preservation of the sphincter muscle is prioritized, neoadjuvant CRT plus consolidation chemotherapy is preferred [34,35,40,41]. Alternatively, if patients have high-risk metastasis, the introduction of chemotherapy plus CRT may be more viable compared with the standard strategy [37,38].

## 5. Neoadjuvant Immunotherapy

Checkpoint immunotherapy (CPI) has achieved great success in the treatment of solid tumors, becoming a new pillar for the treatment of malignant tumors. At present, CPI has seen the most progress in advanced colorectal cancer, primarily in high instability microsatellite (MSI-H) or mismatch repair defect (dMMR) colorectal cancer (Keynote-016, Keynote-164, CheckMate-142, Keynote-177, NICHE) [41,42,43,44,45]. In 2022, an early, small-scale clinical trial of dostarlimab, a blocking monoclonal antibody specific for programmed cell death receptor 1 (PD-1), reported some striking results. Dostarlimab was given to 14 patients with stage II and stage III dMMR rectal adenocarcinoma for six months with the intention of preparing them for standard CRT and surgery [46]. However, within that six-month period, all 14 patients had achieved complete clinical remission. No evidence of tumor presence could be found, as quantified by MRI, fluorodeoxyglucose positron emission tomography (FDG-PET), endoscopy, digital rectal examination, or biopsy. Furthermore, no grade 3–4 adverse events were observed. The results of this study indicated PD-1 inhibitor monotherapy as a radical treatment for LARC with dMMR, one effective enough that recipients might not even require surgery. The follow-up to the study is ongoing, and further efficacy evaluation results are expected.

These results, while promising, must be put into context. Only 2.7% of rectal adenocarcinomas are MMR deficient [47]. Most patients instead show microsatellite stability (MSS) or mismatch repair proficiency (pMMR) status, and these patients exhibit low response rates to immunotherapy [48,49]. Therefore, how to improve the efficacy of immunotherapy for MSS colorectal cancer has become a focus of research.

One method that has proven effective in breaking immunological quiescence and inducing an inflammatory state is to combine immunotherapy with RT (IRT). Radiation can induce inflammatory changes and increase the number and functionality of T cells, NK cells, and antigen-presenting cells through numerous mechanisms such as interferon and Toll-like receptor signaling, induction of immunogenic cell death, and upregulation of cellular stress markers and major histocompatibility complex (MHC) on the tumor cell [50]. Therefore, RT may increase the sensitivity of MSS colorectal cancer to immunotherapy. IRT is expected to achieve better tumor regression and long-term efficacy than either modality alone, and there are, at present several phase II clinical studies of IRT being carried out in LARC.

In the VOLTAGE-A trial from Japan, nivolumab and subsequent radical surgery were performed following preoperative CRT in patients with MSS and MSI-H LARC [51]. Promising pCR rates of 30% and 60%, with mild toxicities, were shown in MSS and MSI-H LARC patients treated with nivolumab plus radical surgery after CRT. In contrast, the pCR of conventional long-course concurrent CRT (LCCRT) was only around 15–20%, suggesting that CRT combined with immunotherapy achieved better short-term efficacy [51] (Table 3). Furthermore, biomarker analysis using tumor biopsy samples showed that patients with tumors that expressed programmed cell death 1 ligand 1 (PD-L1) and who exhibited tumor-infiltrating lymphocytes (TILs) at baseline had higher pCR rates, suggesting that the efficacy of IRT is strongly predicted by the patient’s immune microenvironment.

The Italian ANAVA study included 101 patients with LARC who were followed by six courses of avelumab immunotherapy after CRT. Of the 96 patients who could ultimately be evaluated for pathology, 23% achieved pCR, and 60% achieved a major pathologic response, whereas the rate of grade 3–4 non-immune and immune-related toxicity was only 8% and 4%, respectively [52] (Table 3).

A contrasting result to the above successful trials can be seen in the NRG-GI002 study [53]. In it, all cohorts were treated according to the TNT model. The control group was treated with eight courses of FOLFOX chemotherapy followed by long-course RT, while the study group was treated with eight courses of FOLFOX followed by long-course RT combined with concurrent pembrolizumab. No significant differences were found between the control and study groups. The control group had a mean neoadjuvant rectal cancer (NAR) score of 14.08%, whereas the pembrolizumab had a mean NAR score of 11.53% (*p* = 0.26). The pCR of the control vs. the pembrolizumab group was 29.4% vs. 31.9% (*p* = 0.75), and the cCR was 13.6% vs. 13.9% (*p* = 0.95) (Table 3).

The failure of immunotherapy to further improve tumor regression in this study deserves further consideration. The reason may be that the concurrent administration of CPI and RT resulted in lymphocytes being recruited to the tumor, only to be killed by the irradiation thereof [56]. The administration of immunotherapy following RT may be more reasonable.

Studies have shown that hypofractionated RT has several advantages over conventional fractionated RT: it has less effect on the number of peripheral blood lymphocytes; it inhibits the recruitment of myeloid-derived suppressor cells to tumors; it delivers superior tumor control in mice; and, when combined with immunotherapy, can more reliably and effectively induce tumor control outside of the radiative field [57,58]. The above results provide a theoretical basis for the clinical application of SC-RT combined with immunotherapy.

Recently, a phase II clinical study (NCT04231552) evaluated the efficacy of a triple combination of RT, post-irradiative chemoimmunotherapy, and surgery [54]. Twenty-six patients with pMMR LARC received SC-RT, followed by two courses of CAPOX (capecitabine and oxaliplatin) combined with camrelizumab, and then underwent TME. The pCR rate was as high as 46%, despite roughly half of the patients in this study having at least one high-risk factor (e.g., T4, N2, MRF^+^, a tumor within 5 cm of the anus) [54] (Table 3). This study showed that even in LARC with high-risk factors, SC-RT followed by chemotherapy and immunotherapy achieved improved tumor regression, even when compared with the most intensive TNT regimen currently used in the clinic. In addition, no serious adverse reactions were observed in this study, and all grade 3 hematological toxicities were relieved after treatment. At present, the researchers are preparing to conduct a phase III multicenter randomized control study to follow up on these results.

The Averectal study investigated the efficacy and safety of neoadjuvant SC-RT followed by mFOLFOX6 plus avelumab for LARC [55]. Among the 40 analyzable patients, the pCR rate was 37.5%, significantly higher than the historical control pCR rate (37.5% vs. 16%, *p* = 0.025), and the major pathological response (MPR) rate was 67.5% (Table 3).

Although the above studies are all phase II studies with small samples, their results have consistently shown that RT and chemotherapy combined with immunotherapy achieve good short-term efficacy. In the future, larger-scale clinical studies will be needed to verify these treatments, as well as to more fully catalog their adverse events and long-term survival results. Moreover, the best combination sequence of RT and immunotherapy remains to be defined. Based on current results, however, neoadjuvant CRT combined with immunotherapy appears to be a promising line of therapy for LARC.

## 6. Local Radiotherapy Boosts Treatment

Non-operative modality (NOM) treatment of rectal cancer is an emerging therapy that avoids TME and stoma [59]. For terminally elderly and frail patients who are unfit for surgery or/and chemotherapy, dose escalation is a potentially curative option. The Morpheus trial investigated the role of image-guided adaptive endorectal brachytherapy for patients who had declined TME [60]. Patients with operable cT2-3ab N0 M0 rectal cancer received 45 Gy of pelvic external beam RT (EBRT), divided into 25 fractions, with concurrent 5-FU/capecitabine. Subjects were then randomized to subsequently receive either an EBRT boost of 9 Gy in 5 fractions (Arm A) or three weekly adaptive brachytherapy boosts for a total of 30 Gy in 3 fractions (Arm B). The 2-year survival rate for Arm A was 40.5% vs. 85.1% in Arm B (*p* = 0.001). This trial suggests that these two strategies of radiation dose escalation provide a feasible alternative to surgery and achieve organ preservation in patients with operable rectal cancer.

The OPERA trial evaluated the ability of dose escalation using contact X-ray brachytherapy (CXB) to preserve the rectum in comparison to standard-of-care therapies (EBCRT and TME) [61]. Patients were randomly assigned into one of two arms. Patients in Arm A received EBCRT (45 Gy in 25 factions over 5 weeks) with oral capecitabine (825 mg/m^2^), followed by an EBRT boost of 9 Gy in 5 fractions over 5 days. Patients in Arm B received EBCRT (90 Gy in 3 fractions over 4 weeks) followed by a CXB boost. Between both arms, organ preservation was achieved in 80.5% of patients without compromising their chance of cure. The study results indicate that non-surgical treatment for cT2-cT3a-b rectal cancer is feasible for patients who are otherwise healthy and who wish to avoid surgery (WW). Those who manifest local residual disease or for local regrowth at a later date can be offered salvage surgery immediately.

## 7. Omission of RT

Since the development of TME, RT has not only failed to show a conclusive benefit to OS but is also associated with safety concerns such as higher rates of incontinence, anal mucous loss, anal blood loss, bowel dysfunction, and sexual dysfunction [62]. Given the potential adverse consequences and marginal benefits of neoadjuvant RT, some studies have weighed whether omitting RT from the therapeutic regimen altogether might yield superior outcomes. The previously mentioned FOWARC trial investigated the addition of oxaliplatin with and without preoperative RT [16,17]. The results indicated that the presence or lack of RT did not significantly impact 3-year DFS in patients with LARC. However, patients treated with RT did experience higher toxicity and more postoperative complications. In the CONVERT study, 663 patients with LARC not involving the mesenteric fascia were randomized to receive four cycles of neoadjuvant CAPEOX (nCT) or concurrent chemoradiation (nCRT) with capecitabine alone (no oxaliplatin) [63]. Although the nCRT group had a better degree of tumor regression than the nCT group (38.6% vs. 24.0%, *p* < 0.001), patients in the nCT group exhibited reduced perioperative distant metastasis (0.7% nCT vs. 3.1% nCRT, *p* = 0.034) and rates of prophylactic shunt ileostomy (52.2% nCT vs. 63.6% nCRT, *p* = 0.008). The pCR rate was similar between both groups (11.0% nCT vs. 13.8% nCRT, n.s.), as were sphincter retention and R0 resection rates. However, the results of CONVERT, while having been presented, have not yet been published.

The results of the FOWARC and CONVERT studies both observed an improvement in quality of life, protection of organ function, and better allocation of medical resources in LARC patients who were treated with chemotherapy rather than chemotherapy in conjunction with RT of select patients.

## 8. Watch-and-Wait Treatment Strategy (WW)

TME is associated with a perioperative mortality rate of 1–2%, which increases with age, frailty, and comorbidity. Additionally, it can lead to temporary or permanent colostomy and serious long-term morbidity, such as urinary and sexual dysfunction, in more than 60% of patients [64,65]. In light of these potential complications, the WW strategy was proposed as an alternative by Professor Habr-Gama [66]. In her study, WW was used in LARC patients who experienced cCR after CRT and had good long-term follow-up results. Since then, multiple studies have shown no survival benefit for surgical resection in patients with cCR. The risk of recurrence is higher than that of standard treatment. However, these results have been widely questioned. Studies have shown that if the clinical evaluation is cCR, only 36% of patients with real pCR after direct surgery. Furthermore, international consensus has not been reached on imaging strategies and timing to identify a cCR, nor have standardized follow-up protocols for timely detection of tumor regrowth been established. The timing of neoadjuvant therapy and the selection of CRT dose vary greatly in various studies, resulting in a wide range of cCR rates (10–78%).

In order to address this shortfall, an international Observation and Waiting Database was established. The results of the database were reported in The Lancet in 2018 [67]. Complete CR was reported in 87% of patients in the database. The median follow-up time was 3 years. The 2-year cumulative incidence of local regrowth was 25.2%; 88% of all local regrowth was diagnosed in the first 2 years, and 97% of local regrowth was located in the bowel wall. Distant metastasis was diagnosed in 8% of 880 patients. Five-year OS was 85%, and 5-year disease-specific survival was 94%.

The OPRA trial (not to be confused with the OPERA trial mentioned previously) investigated the safety and efficacy of WW in the two different TNT models. Patients with MRI stage II and III rectal cancer were randomized to four months of FOLFOX or CAPEOX before (induction) or after (consolidation) fluorouracil or capecitabine-based chemoradiotherapy (CRT). Patients were then re-staged 8–12 weeks after finishing TNT with a digital rectal exam, flexible sigmoidoscopy, and MRI [36,68]. The results, like those of the CAO/ARO/AIO-12 study, demonstrated that consolidation TNT had a higher rate of organ preservation [33,34] (Table 2).

A growing number of studies have explored different modalities of neoadjuvant therapy to improve pCR and cCR rates, enabling the “WW” model to be applied. Altogether, the evidence so far indicates that WW is potentially feasible, but prospective clinical studies are still needed.

## 9. Local Excision

Large databases show that 97% of recurrent sites are confined to the intestinal wall. If the cancer is still aggressive, it has generally been considered preferable to perform a local excision to remove the recurrence.

GRECCAR 2 was the first multicenter, randomized trial to compare local excision with TME in downstaged low rectal cancer [69,70]. The study failed to show the superiority of local excision over TME, either at the 3-year or 5-year marks, as many patients in the local excision group received completion TME that likely increased morbidity and side effects and compromised the potential advantages of local excision.

Despite this initial failure, two subsequent studies would later demonstrate local excision as a viable alternative to TME. The TAUTEM study aimed to compare local recurrence at two years in patients treated with preoperative CRT and transanal endoscopic microsurgery (TEM) with patients undergoing conventional radical TME [71]. The study showed that nCRT-TEM treatment achieved high pathological complete response rates (44.3%), with a high CRT compliance rate (98.8%) and low morbidity. Postoperative complications and hospitalization were also significantly lower in the CRT-TEM group.

The STAR-TREC trial compared non-operative organ preservation (OP) therapy for early-stage rectal cancer versus standard-of-care using TME alone [72,73]. Patients were randomly assigned to one of three groups: TME, OP via mesorectal SC-RT (5 × 5 Gy), or OP via mesorectal CRT (25 × 2 Gy + capecitabine). Standardized response assessment classified OP cases that required no further treatment after 20 weeks as a complete response, that subsequently received transanal endoscopic microsurgery within 20 weeks as a partial response, and that subsequently received TME within 20 weeks as a poor response. Surveillance following OP consisted of three monthly endoscopies/MRIs. In addition, all patients received thoracic, abdominal, and pelvic CT scans at 24 months. Both OP methods were found to reduce acute surgical morbidity without introducing substantial radiation toxicity. While the OP methods exhibited a reduced 24-month DFS compared to TME (75.1% vs. 91.2%), there was no significant difference in non-regrowth DFS at 24 months (90.1% vs. 85.9%). The overall quality of life was evenly matched. STAR-TREC phase III will determine the optimal strategy for achieving OP. STAR-TREC’s results have been presented but not yet published. We are looking forward to the full report.

Together, the results of TAUTEM and STAR-TREC show that, when paired properly with RT/CRT, local resection of the tumor can achieve results comparable to TME while preserving the rectal organ.

## 10. Conclusions

In the era of precision medicine era, final treatment decisions should be based on risk stratification. Not all LARC cases are appropriate for the predominant neoadjuvant “sandwich” treatment. For patients with low and medium risk of recurrence, less intensive regimens may yield equivalent disease responses while also greatly preserving their quality of life. For patients at high risk for recurrence, TNT is emerging as an ideal strategy. Future studies will help to further establish risk stratification groups and clarify the ideal regimen and optimal sequencing of chemotherapy and radiation in the setting of TNT. ACO/ARO/AIO-18.1 (ClinicalTrials.gov Identifier: NCT04246684) is an ongoing randomized trial that aims to directly compare the efficiency of SC-RT-based consolidation TNT according to RAPIDO with LCRT-based consolidation TNT according to CAO/ARO/AIO-12. The results of this trial will provide evidence to establish the optimal regimen of TNT. Local resection and X-ray proximal boost irradiation also provide a promising new protocol for organ preservation. Furthermore, some studies have achieved pCR rates of over 30%, enabling the “WW” model to be applied. Finally, CRT combined with immunotherapy has achieved good short-term efficacy. This treatment mode is highly worthy of further exploration. The results of the ongoing phase II/III STELLAR-2 trial (ClinicalTrials.gov Identifier: NCT05484024) will inform as to the efficiency of sequential neoadjuvant SC-RT and chemotherapy with PD-1 inhibition.

## Figures and Tables

**Table 1 cancers-15-00219-t001:** Intensifying concurrent chemotherapy regimen.

Trials	Patients (*n*)	Treatment Methods		Results (Con vs. Exp)	Conclusion	Ref
		RT	Concurrent Chemotherapy			
CAO/ARO/AIO-04	Con = 623	50.4 Gy(5 × 1.8 Gy/w)	Con: 5-FU 1000 mg/m^2^/d d1–5, d29–33	pCR: 13% vs. 17%, *p* = 0.031	Adding oxaliplatin significantly improved DFS and pCR in patients.	[12,18]
	Exp = 613		Exp: 5-FU 250 mg/m^2^/d d1–14, d22–35 Oxaliplatin 50 mg/m²/d d1, 8, 22, 29	3 y-DFS: 71.2% vs. 75.9%, *p* = 0.03		
				Preoperative grade 3–4 adverse events: 20% vs. 24%, ns		
ACCORD12	Con = 299	Con: 45 Gy (5 × 1.8 Gy/w)	Con: Capecitabine 800 mg/m^2^ × 2/d 5 d/w	ypCR: 13.9% vs. 19.2%, *p* = 0.09	The benefit of oxaliplatin was not demonstrated.	[11]
	Exp = 299	Exp: 50 Gy(5 × 2 Gy/w)	Exp: Capecitabine 800 mg/m^2^ × 2/d 5 d/w Oxaliplatin 50 mg/m^2^/w qw	Preoperative grade 3 to 4 adverse events: 11% vs. 25%, *p* < 0.001		
STAR-01	Con = 379	50.4 Gy(5 × 1.8 Gy/w)	Con: FU 225 mg/m^2^/d	pCR: 16% vs. 16%, *p* = 0.904	Adding oxaliplatin significantly increases toxicity without affecting primary tumor response.	[10]
	Exp = 368		Exp: 5-FU 225 mg/m^2^/d Oxaliplatin 60 mg/m^2^/w × 6 w	Preoperative grade 3 to 4 adverse events: 8% vs. 24%, *p* < 0.001		
NSBP R-04	Con = 949	45 Gy or 50.4 Gy or 55.8 Gy (5 × 1.8 Gy/w)	Con: 5-FU 225 mg/m^2^/d 5 d/w or Capecitabine 825 mg/m^2^ × 2/d 5 d/w	pCR:17.8% vs. 19.5%, *p* = 0.42	Adding oxaliplatin did not improve surgical outcomes but added significant toxicity.	[13]
	Exp = 659		Exp: 5-FU 225 mg/m^2^/d 5 d/w or Capecitabine 825 mg/m^2^ × 2/d 5 d/w Oxaliplatin 50 mg/m^2^/w × 5 w	Grade 3 to 5 adverse events: 6.9% vs. 16.5%, *p* < 0.001		
PETACC 6	Con = 543	45 Gy or 50.4 Gy (5 × 1.8 Gy/w)	Con: Capecitabine 2 × 825 mg/m^2^ × 2/d	pCR: 11.6% vs. 14.0%, *p* = 0.225	The addition of oxaliplatin to preoperative capecitabine-based chemoradiation and postoperative adjuvant chemotherapy impaired tolerability and feasibility without improving efficacy.	[14]
	Exp = 525		Exp: Capecitabine 2 × 825 mg/m^2^ × 2/d Oxaliplatin 50 mg/m^2^/d d1, 8, 15, 22, 29	7 y-DFS: 66.1% vs. 65.5%, *p* = 0.861		
				7 y-OS: 73.5% vs. 73.7%, *p* = 0.205		
FOWARC	Con = 155	Con and Exp_1_: 46.0 Gy (5 × 2 Gy/w) or 50.4 Gy (5 × 1.8 Gy/w)	Con: (Leucovorin 400 mg/m^2^ + 5-FU 400 mg/m^2^ + 5-FU 2.4 g/m^2^ d1–2/2 w)× 5 cycles	pCR: Con vs. Exp_1_: 14.0% vs. 27.5%, *p* = 0.005	mFOLFOX6-based preoperative chemoradiotherapy results in a higher pCR rate than 5-FU-based treatment but did not significantly improve 3 y-DFS.	[16,17]
	Exp_1_ = 157		Exp_1_ and Exp_2_: (Leucovorin 400 mg/m^2^ + 5-FU 400 mg/m^2^ + 5-FU 2.4 g/m^2^ d1–2/2 w + oxaliplatin 85 mg/m^2^/2 w)× 5 cycles	3 y-DFS: Con vs. Exp_1_ vs. Exp_2_: 72.9% vs. 77.2% vs. 73.5%, *p* = 0.709		
	Exp_2_ = 163			3 y-OS: Con vs. Exp_1_ vs. Exp_2_: 91.3% vs. 89.1% vs. 90.7%, *p* = 0.971		
CinClare	Con = 178	50 Gy (5 × 2 Gy/w)	Con: Capecitabine 825 mg/m^2^ × 2/d 5 d/w Oxaliplatin 130 mg/m^2^ d1 Capecitabine 1000 mg/m^2^ × 2/d d1–14	pCR: 15% vs. 30%, *p* = 0.001	Adding irinotecan guided by the UGT1A1 genotype to capecitabine-based neoadjuvant chemoradiotherapy significantly increased complete tumor response in Chinese patients.	[19]
	Exp = 178		Exp: Capecitabine 625 mg/m^2^ × 2/d 5 d/w Irinotecan UGT1A1*1*1, 80 mg/m^2^ /w UGT1A1*1*28, 65 mg/m^2^/w Irinotecan 200 mg/m^2^ d1 Capecitabine 1000 mg/m^2^ 2/d d1–14	Grade 3–4 adverse events: 6% vs. 38%, *p* < 0.001		
ARISTOTLE	Con = 284	45 Gy (5 × 1.8 Gy/w)	Con: Capecitabine 900 mg/m^2^ × 2/d	pCR: 17.4% vs. 20.2%, *p* = 0.45	The addition of irinotecan did not significantly improve the pCR rate and was associated with a decrease in the RT and capecitabine compliance and a higher rate of adverse events.	[20]
	Exp = 280		Exp: Capecitabine 650 mg/m^2^ × 2/d Irinotecan 60 mg/m^2^ /w × 4 w	Grade 3–4 adverse events: 12% vs. 21%, *p* = 0.004		

Exp = Experimental; Exp_1_ = Experimental-1; Exp_2_ = Experimental-2; Con = Control; DFS = disease-free survival; Gy = Gray; d = day; w = week; y = year; RT = Radiotherapy; pCR = pathological complete response; * is part of the genotyping nomenclature.

**Table 2 cancers-15-00219-t002:** Total neoadjuvant therapy.

Trials	Patients (*n*)	Treatment Methods	Results(INCT vs. CNCT)	Conclusion	Ref
CAO/ARO/ AIO-12	INCT = 156	INCT: chemotherapy/CRT/surgery	pCR: 17% vs. 25%	CNCT resulted in better compliance with CRT but worse compliance with chemotherapy compared with INCT.	[33,34]
			3 y-DFS: 73% vs. 73%, *p =* 0.82		
	CNCT = 150		OS: 92% vs. 92%, *p =* 0.81		
			CRT-related grade 3 or 4 toxicity: 37% vs. 27%		
OPRA	INCT = 152		DFS: 78% vs. 77%, *p =* 0.90	CNCT resulted in a numerically higher WW rate compared to induction chemotherapy followed by CRT.	[36]
	CNCT = 155	CNCT: CRT/chemotherapy/surgery	DMFS: 81% vs. 83%, *p =* 0.86		
			OP: 43% vs. 58%, *p =* 0.01		
PRODIGE 23	INCT = 230		3 y-DFS: 69% vs. 76%, *p =* 0.034	Neoadjuvant mFOLFIRINOX plus CRT is safe and significantly increases ypCR rate, DFS and MFS.	[37,38]
			3 y-MFS: 71.7% vs. 78.8%, *p <* 0.02		
	CNCT = 231		pCR: 11.7% vs. 27.5%, *p <* 0.001		
			3 y-OS: 87.7% vs. 90.8%, *p =* 0.077		
RAPIDO	standard arm = 441	standard arm: RT: 50 Gy (5 × 2 Gy/w) or 50.4 Gy (5 × 1.8 Gy/w)	3 y-DrTF: 30.4% vs. 23.7%, *p =* 0.019	The 3 y-DrTF rate was significantly reduced by 7%, and the pCR rate was increased from 14% to 28% in the short-course radiotherapy, followed by consolidation chemotherapy and TME.	[39]
	experimental arm = 460	capecitabine experimental arm: RT: 5 × 5 Gy CAPOX × 6 cycles or FOLFOX4 × 9 cycles	3 y-OS: 88.8% vs. 89.1%, *p =* 0.59		
			R0 resection rate: 90% vs. 90%, *p* = 0.87		
			pCR: 14% vs. 28%, *p* < 0.0001		

INCT = induction chemotherapy; CNCT = consolidation chemotherapy; CRT = chemoradiotherapy; DMFS = distant metastasis-free survival; OP = organ preservation; DrTF = disease-related treatment failure; Gy = Gray; TME = total mesorectal excision; WW = watch and wait; y = year.

**Table 3 cancers-15-00219-t003:** Neoadjuvant immunotherapy.

Trials	Patients (*n*)	Patients and Methods		Results	Conclusion	Ref
		RT	Chemotherapy and Immunotherapy			
VOLTAGE-A	A1(MSS) = 37	50.4 Gy	Capecitabine + nivolumab	A1: pCR: 30% (11/37) mpR: 38% (14/37)	Promising pCR rates of 30% and 60%, with mild toxicities, were shown in MSS and MSI-H LARC patients treated with Nivolumab plus radical surgery after CRT, suggesting the candidate therapy for the future non-surgical approach.	[51]
	A2(MSI-H) = 5			A2: pCR: 60% (3/5) mpR: 60% (3/5)		
ANAVA	101	50.4 Gy	Capecitabine + avelumab	pCR: 23% (22/96)	The combination of preop CRT plus avelumab showed promising activity and a feasible safety profile.	[52]
				grade 3–4 non-immune adverse events: 8%		
				grade 3–4 immune-related adverse events: 4%		
NRG-GI002	Con = 95	50.4 Gy	Con: FOLFOX + capecitabine Exp: FOLFOX + capecitabine + pembrolizumab	Mean NAR: con vs. exp =14.08 vs. 11.53 *p* = 0.26	Pembrolizumab added to CRT as part of TNT was safe and without unexpected short-term toxicities but failed to improve the NAR score.	[53]
				pCR: 29.4% vs. 31.9%, *p* = 0.75		
	Exp = 90			cCR: 13.6% vs. 13.9%, *p* = 0.95		
				SSS: 71.0% vs. 59.4%, *p* = 0.15		
NCT04231552	30	25 Gy	Oxaliplatin + capecitabine + camrelizumab	pCR: 48.1% (13/27)	SC-RT combined with subsequent CAPOX plus camrelizumab followed by delayed surgery showed a favorable pCR rate with good tolerance in patients with LARC, especially in the proficient MMR setting.	[54]
				pMMR/MSS: 46% (12/26)		
				dMMR/MSI-H: 100% (1/1)		
Averectal	44	25 Gy	mFOLFOX6 + avelumab	pCR: 37.5% (15/40) mpR: 67.5% (27/40)	The primary endpoint was successfully met with significant improvement in pCR and mpR rates in the setting of an acceptable safety profile.	[55]

cCR = clinical complete response; Con = Control; CRT = chemoradiotherapy; Exp = Experimental; Gy = Gray; LARC = locally advanced rectal carcinoma; MMR = mismatch repair; mpR = major pathological response; NAR = neoadjuvant rectal cancer; pCR = pathological complete response; SC-RT = short course radiotherapy; SSS = sphincter-sparing surgery; TNT = total neoadjuvant therapy; A1 = cohort A-1; A2 = cohort A-2; MSS = microsatellite stability; MSI-H = high instability microsatellite.

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
