# Peer review of "Strategies to Optimize Treatment for Locally Advanced Rectal Cancer"

_cancers, 2022, doi:10.3390/cancers15010219_

Round 1
Reviewer 1 Report
1: Abstract looks incomplete, which is ending up with objectives only, please provide a brief summary of methods, results and conclusion.
2: Introduction describe the rationale for the review in the context of existing knowledge and provides an explicit statement of objectives.
3: Methods: I have few comments on methods.
It must be explained under a separate heading of methods.
Several clinical trials were selected but it is unclear what was the inclusion and exclusion criteria while selecting these trials. What methods were specified to see if the study is meeting the inclusion criteria.
What search engines, data bases were used to ensure no important clinical trials were missed out as per review objectives or clinical questions.
4: Throughout the manuscrtir authors are referring ‘several trials have been undertaken” however not a signle refrences is provided, please check and add suitable refrences to support the statement. One of the example is:
Page 6: 3. Total neoadjuvant therapy
Several trials have been undertaken to evaluate the role of adjuvant chemotherapy
after neoadjuvant CRT/short course RT (SC-RT) and surgery.
5: Registration information is missing, please provide if this review was registered
Reviewer 2 Report
Dear Authors,
I congratulate you for writing the review. However, I have some comments that I hope you can address.
Point by point comment:
- Abstract
o „followed with optional chemotherapy“
§ Better: followed by optional adjuvant chemotherapy
o “For oncology effect”
§ Do you mean outcome? – Then maybe outcome is the better word.
- Introduction
o “Due to development of magnetic resonance imaging”
§ How can MRI development improve the rate of local recurrence rates in a clinical cohort?
o “anal preservation”
§ I would recommend using the term sphincter preservation.
o “TMRES”
§ Do you mean TME?
o “intensive neo-CRT”
§ Do you mean total neoadjuvant treatment strategies?
- Intensifying concurrent chemotherapy regimen
o “local regional recurrence rates ..” than you write “better control of systemic disease”
§ Not local regional recurrence correlated with DFS and OS today but the much higher rate of distant metastasis have relevant impact on survival.
o “oxaliplatin was administered on a weekly basis”
§ That is not correct for the CAO/ARO/AIO-04 trial – Oxaliplatin was administered on day 1,8,22,29 – on day 15 was a pause
o “Association of Treatment Adherence With Oncologic Outcomes for Patients With Rectal Cancer: A Post Hoc Analysis of the CAO/ARO/AIO-04 Phase 3 Randomized Clinical Trial”
-> This paper proposed that treatment adherence is crucial for the benefit of intensified nCRT. You may can included this aspect in your review.
o “Prodige 23 trial adopted total neoadjuvent therapy (TNT”)
§ You introduce the term TNT without any explanations! Please explain why TNT was investigated (e.g. worse adherence to adjuvant CRT-hypothesis that adherence could be better if CT was conducted before surgery,..)
o Why is that in part 3? Not in the TNT part?
- Total neoadjuvant treatment
o “from these studie two TNT paradigmes have emerged”
§ Maybe it is better to write that two TNT paradigms have been investigated in the recent trials …
o “that up-front CT followed by consolidation is the preferred”
§ Be careful. You have to comment here on the time aspect. Tumor cells in patients with consolidation chemotherapy have more time to shrink and die after radiotherapy compared to patient in the induction chemotherapy arm. This aspect have to be dicussed!
§ “Total Neoadjuvant Therapy for Rectal Cancer in the CAO/ARO/AIO-12 Randomized Phase 2 Trial: Early Surrogate Endpoints Revisited” is maybe helpful
o This abstract is to short:
Please comment on the RAPIDO as landmark TNT trial. Please comment on the higher rate of localrecurrence in the Rapido trial - "Patterns of locoregional failure and distant metastases in patients treated for locally advanced rectal cancer in the Rapido trial."
o “efficacy of WW”
§ Again you introduce the aspect of WW without any IntroductIonal remarks. Please explain what W&W in the clinical routine means.
o “Introduction chemotherapy plus CRT may be more viable”
§ Is that your statement our can you cite clinical studies which support your “statement”
- Neoadjuvant immunotherapy
o “only appears in 15-20% of stage II/III”
§ There was a NEJM comment “Prevalence of Mismatch-Repair Deficiency in Rectal Adenocarcinomas”. They report only a 2.7% MMR deficient rate. You should refer to this number because it brings the Dostarlimab trial in better context for the overall treatment of locally advanced rectal cancer.
o You should mentioned the low number of patients in all immune trials.
o “clinical applicaton of SC-RT”
§ Again, you introduce the term short course radiotherapy without introduction.
è Maybe it would be better to introduce all treatment modalities you reviewed in the introduction section.
- Local radiotherapy boost treatment
o You may add the termin eldery and frail. Because for these patients who are unfit for surgery or/and chemotherapy dose escalation is a potentially curative option.
- Short-course radiotherapy
o I would recommend to review short course radiotherapy before the TNT section.
o Rapido is a TNT trial!
o At a median follow-up of 35.0 months, 3-year DFS was 64.5% and 62.3% in TNT and CRT groups, respectively (hazard ratio, 0.883; one-sided 95% CI, not applicable to 1.11; P < .001 for noninferiority)
§ -> That is from the Stellar trial Abstract in JCO. The P value is the result of a noninferiority test and not of a log rank test. Please correct your statement.
- Omission of RT
o Convert trial
§ The trial has only been published as abstract so far, right? Please mention this limitation.
§ nCT reduced perioperative distant metastases ! – The sentence “exhibit increased metastasis” is misleading without the information that we are talking about perioperative distant metastases.
§ “less preventive ileostomy” -> The trial have to comment on the localization of the tumor and their distribution between both arm. If the trial have not reported that have to be mentioned as limitation
- Watch and Wait
o The OPRA trial should be comment in this section !
- Local Excision
o Star Trec only published as abstract. Please mention this limitation.
- Conclusion
o Please add information on some ongoing phase III trials
o e.g. Clinicaltrial.gov rectal cancer phase III trials
Overall remarks
As mentioned above you should describe the categorization of your review within your Introduction. Perhaps you should reconsider your thematic order.
Proposal:
1. Long-course CRT vs short-course RT
2. Intensified CRT
3. TNT
4. Watch & Wait
5. RT concepts with e.g. Brachy Boost
6. W&W
7. Immunotherapy
8. Omission of RT
Reviewer 3 Report
The work is a highly comprehensive structured study of currently used locally advanced rectal cancer optimization strategies. It contains an analysis of data from over 60 source publications. The individual aspects described by the authors are grouped into well-prepared chapters related to a very well-constructed narrative. The positive image of the work is complemented by well-composed tables that provide condensed information. Apart from the incorrect formatting of citations, this work cannot be faulted.
Author Response
We are very grateful for your comments on the manuscript. To further improve the manuscript, we have made some amendments in several parts. And the amendments are highlighted in the revised manuscript.
Round 2
Reviewer 2 Report
Thanks for considering the inital comments.
Just some minor recommendation/comments:
"European Society for Medical Oncology (ESMO) guidelines can classify the risk of recurrence as low to very high based on magnetic resonance imaging (MRI)"
-> Better: According to ESMO guidelines risk of recurrence can be classified by MRI as low to very high
"Multiple clinical trials focused on optimizing strategies to acquire oncology effectoutcomes and functions for LARC"
-> to improve outcome [not to acquire]
"aiming to maximize tumor regression for organ function preservation, including intensity concurrent chemotherapy"
-> intensified not intensity
"There are two ongoing phase III trials to explore whether SC-RT addplus chemotherapy is superior to LCRT: the STELLAR and Polish II trial"
-> Both trials are published therefore "ongoing" might not be correct here.
"Thus, although the exact parameters in which benefit was observed differed between the two studies, both found that SC-RT yielded superior results to LCRT."
-> Better: that SC-RT combined with consolidation CT is not inferior to LCRT
"The results of the FOWARC and CONVERT studies both indicate that the use of chemotherapy alone, rather than in conjunction with RT, may significantly improve posttreatment quality of life of patients with LARC, protecting organ function and optimizing the allocation of medical resources"
-> better: Improvement of the quality ... of selected patients ...
